**Biological science practices**  

evolution

preprint, early-career scientists, peer review, *Proceedings B*, Royal Society of London

**Author for correspondence:**
Maurine Neiman
e-mail: maurine-neiman@uiowa.edu

# Development, implementation and impact of a new preprint solicitation process at *Proceedings B*

Maurine Neiman[1,2], Robin K. Bagley[3], Dorota Paczesniak[4] and Shalene Singh-Shepherd[5]

[1]Department of Biology, and [2]Department of Gender, Women's, and Sexuality Studies, University of Iowa, Iowa City, IA 52242, USA
[3]Department of Evolution, Ecology, and Organismal Biology, The Ohio State University at Lima, Lima, OH 45804, USA
[4]Department of Ecology and Genetics, University of Oulu, Oulu 90014, Finland
[5]The Royal Society, 6-9 Carlton House Terrace, London SW1Y 5AG, UK

MN, 0000-0002-1543-8115; RKB, 0000-0003-2209-0521; SS-S, 0000-0002-7494-7788

Preprints are manuscripts posted on a public server that do not yet have formal certification of peer review from a scholarly journal. The increasingly prominent online repositories for these preprints provide a means of rapidly making scientific results accessible to all with an Internet connection. We here describe the catalysis and subsequent development of a successful new process to solicit preprints for consideration for publication in *Proceedings B*. We present preliminary comparisons between the focal topics and geographic origin of submitting authors of papers submitted in the traditional (non-solicited) route versus solicited preprints. This analysis suggests that the solicitation process seems to be achieving one of the primary goals of the preprint solicitation endeavour: broadening the scope of the papers featured in *Proceedings B*. We also use an informal survey of the early-career scientists that are or have been involved with the Preprint Editorial Team to find that these scientists view their participation positively with respect to career development and knowledge in their field. The inclusion of early-career researchers from across the world in the preprint solicitation process could also translate into social justice benefits by providing a career-building opportunity and a window into the publishing process for young scientists.

## 1. Introduction

Preprints are manuscripts posted on a public server that do not yet—and might never [1]—have the formal certification of peer review from a scholarly journal. Each preprint is assigned a unique and permanent digital object identifier, meaning that preprints present a time-stamped and citable contribution to knowledge. Preprints allow scholars to make research findings freely and immediately available to a global audience, providing a partial solution to ongoing challenges associated with peer review such as delays in publication [2] and reviewer bias ([3,4]; reviewed in [5–7]). The open-access component of preprints is in itself a positive: open-access articles are recognized earlier and cited more often than non-open-access counterparts, thus accelerating the rate at which new scientific results are shared and used [8]. Early-career scientists can be particular beneficiaries of a culture that welcomes and values preprints, which can provide these scientists a mechanism to rapidly disseminate their work, establish precedence, and build a wider reputation than might be possible via traditional publication channels [5,7,9]. From a social justice perspective, preprints provide a means of sharing science that defies

traditional cultural, social, and economic barriers, though formidable challenges regarding equitable access, promotion, and recognition remain [10].

Although preprints only recently rose to prominence, they were first introduced in 1961 as part of a US National Institutes of Health project called the Information Exchange Groups. The widespread acceptance of preprints required global availability of the Internet, the concomitant maturation of information technology, and willingness of journals to accept papers originally posted as preprints (reviewed in [6,10]).

There is an increasing sense that preprints constitute meaningful scholarly contributions [10,11], exemplified by a growing list of prestigious national agencies and funding bodies (e.g. National Institutes of Health, the European Research Council, Centre National de la Recherche Scientifique (CNRS) France, Wellcome Trust) that encourage researchers to cite preprints in grant applications and as evidence of productivity from a grant (ASAPbio, https://asapbio.org/funder-policies; [10]). In recent years, some researchers even publish preprints as a 'final product', with no intention to submit for formal peer review and publication [1].

## 2. The road to a preprint editor at *Proceedings B*

Aware of the growing prominence of preprints and that other publishers were starting to screen preprint servers for appropriate manuscripts, the Editorial Board of *Proceedings B* decided in 2017 to appoint Maurine Neiman as Preprint Editor from the slate of Associate Editors to take on such a role. In particular, *Proceedings B* was hoping to use the recruitment of submissions from preprint servers to increase disciplinary breadth, especially with respect to papers outside of the behaviour-, evolution-, ecology-, and organismal biology-focused research that often makes up the bulk of their submissions. The editors were also hoping to raise awareness of the journal across biology. Finally, the Editorial Board was interested in the potential of preprint solicitation to increase participation and inclusion of perspectives of authors from a wider range of countries than currently featured in *Proceedings B*.

### (a) Developing the process

We started by focusing on preprints posted in *bioRχiv* because this fairly new server had been modelled after a*Rχiv*, one of the original and most successful preprint servers. *bioRχiv*'s search settings also enabled structured scans for papers within particular time frames and biological subdisciplines. The latter mapped nicely onto *Proceedings B*'s disciplinary coverage, which extends into biomedicine, bioengineering and biophysics, psychology, and epidemiology. Finally, *bioRχiv* is having an impact on our field: nearly 40 000 preprints were uploaded to *bioRχiv* in its first 5 years, and the number of submissions per year is steadily increasing [12]. Preprints deposited in *bioRχiv* are also cited more often and have higher Altmetric ratings than counterparts that were not deposited in *bioRχiv* [13].

The disciplinary breadth of *Proceedings B* meant that we would need a broadly trained team to identify potentially appropriate papers across biology. Neiman was fortunate to have such resources available in the University of Iowa graduate and undergraduate students and postdocs in her courses, laboratory group, and across campus. An editorial team made up of early-career researchers also afforded a unique opportunity to empower young scientists by providing a direct opportunity to interact with the scientific reviewing and publishing process. Neiman enlisted about 20 of these early-career scientists, drawing from expertise in biology and beyond, to serve on the brand new Preprint Editorial Team.

Together, the group formulated their process (figure 1, for overview). After several rounds of solicitations and rigorous troubleshooting, we came up with a system that allowed us to efficiently survey thousands of papers from across biology each month. First, team members are assigned a particular subject area in *bioRχiv* that also fits within *Proceedings B*'s purview. Each team member then scans through some or all (depending on the total number of submissions) of the manuscripts submitted in that subject on a monthly basis. We request that new team members familiarize themselves with *Proceedings B*'s scope in their focal subject area by reviewing recently published papers in *Proceedings B* in their assigned area of focus and scanning preprints that other members of the topical team have suggested in the past few months. We also reminded team members that part of our task was to increase the breadth of submissions to *Proceedings B*, so that we were especially interested in identifying preprints that fit the topical purview of *Proceedings B* but that might focus on topics not often seen in the journal.

The dozens to even hundreds of preprints that each team member is responsible for each month (figure 5) meant that we needed to implement mechanisms that allowed us to quickly make decisions on each preprint. First, we recommended that team members use titles to exclude preprints that have a narrow scope or are limited to methodological developments. Next, team members skim the abstract and, often, the full text of those preprints that pass the initial title screen to get a sense of what questions the paper is asking and how findings are presented. In particular, we are looking for papers that address questions likely to be of interest to biologists from multiple subdisciplines and for papers that frame these questions in a broader biological context. Identifying information for appropriate papers is then added to a cloud-based spreadsheet that all team members can access and edit. In general, the articles and the preprints that we solicit for submission are those that we believe would be of broad interest to biologists as a whole, though some of the preprints will need substantial editing (often, additional text in the introduction and/or discussion) to make their breadth clear.

Neiman took on the responsibility of checking each of these prospective solicitations and making a final decision regarding whether to email the corresponding author to encourage a submission. Neiman also provided specific feedback to team members regarding the reasons for choosing not to solicit particular preprints (e.g. too narrowly focused, out of scope, too long). We recommend that team members use this feedback to refine their preprint selection process. As time went on and the team grew, Neiman appointed team members (and co-authors) Dr Robin Bagley and later Dr Dorota Paczesniak as 'Associate Editors', tasked with handling communications, subject area assignments, and recruitment. We also used monthly in-person meetings to discuss our process and to suggest improvements. Within the last year, we have also streamlined the submission process for solicited preprints, enabling direct access to submission to *Proceedings B* from within *bioRχiv*.

Proc. R. Soc. B 288: 20211248

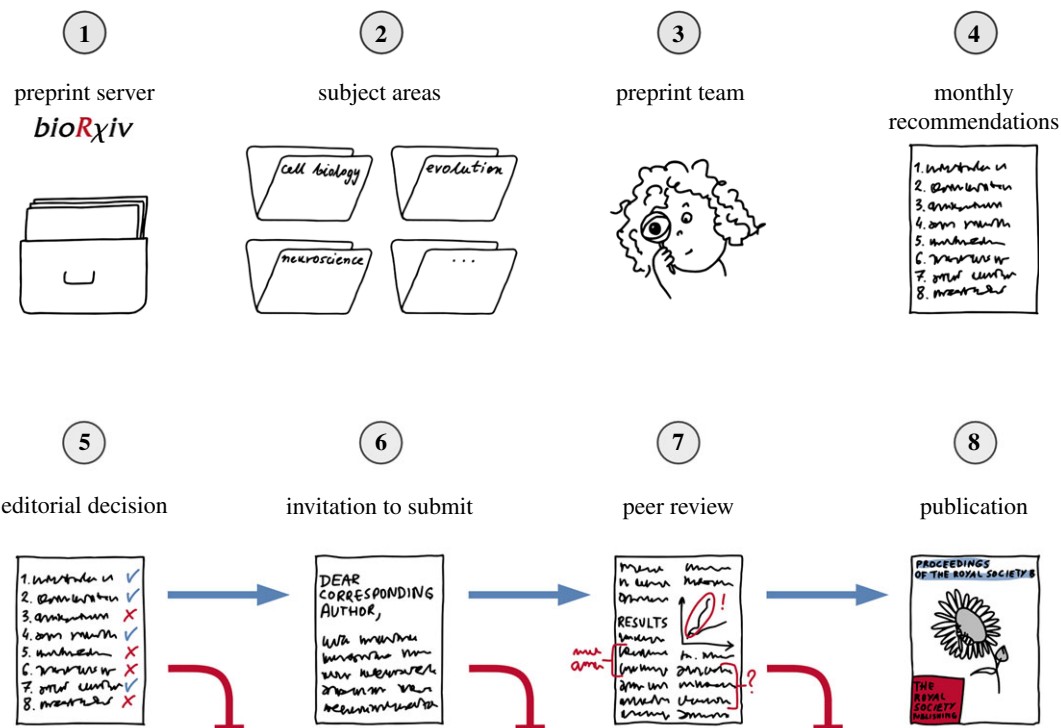

**Figure 1.** Overview of the preprint screening and solicitation process. Preprints from the *bioRχiv* server (1) are screened by subject area (2) by team members (3), who then submit their monthly recommendations (4). Decision whether to solicit recommendations is taken by the Preprint Editor (5), and invitations are sent to corresponding authors (6). If authors decide to submit their work to be considered for publication at *Proceedings B*, the manuscripts go through a standard peer review process (7), before eventual publication in the journal (8). (Online version in colour.)

While a few authors that we have solicited have responded negatively (all regarding the solicitation as spam), our general sense is that our efforts are welcome: we have received hundreds of positive responses since we began this project, ranging from simple *'thank you for your interest'* replies to heartening messages along the lines of *'I hadn't thought about Proceedings B until I received your email, and I think it could be a great fit'* to encouraging words about our use of preprint servers. These responses also revealed another phenomenon: the majority of papers that we solicit for submission are already in review. What this meant is that many authors responded by saying that they would either consider *Proceedings B* if their paper was rejected or submit to *Proceedings B* in the future. This information also provided a good explanation for our unanimous perspective that the papers posted on *bioRχiv* are almost always of very high quality with respect to formatting, structure, and visual elements.

By 2019, transitions like graduations or new positions meant that most of the early-career trainees that were part of the first iteration of the Preprint Editorial Team had moved on. In response, we initiated an international search for new team members, using *Proceedings B*'s Twitter account and website and our own personal social media accounts to announce the opportunity. We ran another such search in 2020. These searches have been successful, and we have to date recruited over 60 early-career scientists from 29 institutions in 10 countries across the world (e.g. Canada, Italy, Brazil, and New Zealand). We are grateful to have been able to attract team members from across the world, but we recognize that additional team members representing even more geographical diversity and a greater breadth of perspectives would be an improvement. We will continue to make an effort to recruit as broadly as possible via social media, the *Proceedings B* website, and our own personal networks.

## 3. Are we meeting our goals?

As of January 2021, we have sent 1469 personalized solicitation emails to individual authors regarding 1239 manuscripts (accounting for the 230 solicited manuscripts that listed greater than one corresponding author; figure 2). Informal feedback from solicited preprint authors as well as on social media, in person, and in peer-reviewed literature [10] suggests that we are accomplishing our goals of raising awareness of *Proceedings B*, especially among researchers who might not otherwise think to submit to this journal. We also hope that this means that we are making a good case for preprint servers as a mechanism of making science more widely accessible and increasing scholarly productivity.

We used data on acceptance rates and topical focus to take first steps towards addressing the more concrete goal of increasing the disciplinary breadth of papers and geographic representation of the authors of papers submitted to *Proceedings B* (electronic supplementary material, tables S1 and S2). As of January 2021, 96 solicited articles had been submitted to *Proceedings B* for consideration since we started soliciting preprints in September 2017. Twenty-nine of those articles (approx. 30%) were ultimately accepted for publication (figure 2). We used a Fisher's exact test (as implemented at https://www.socscistatistics.com/tests/fisher/default2.aspx) to compare the number of accepted versus non-accepted papers for each submission type. This analysis revealed that a significantly higher proportion of solicited papers were accepted for publication than papers submitted through the 'traditional' (non-solicited) route in 2017–2020 (2445 accepted/11 583 submitted; approximately 21%) ($p = 0.0332$, figure 2).

We then used a Wilcoxon signed-ranks test (as implemented in IBM SPSS Statistics v. 27.0) comparing representation

**Figure 2.** (*a*) Overview of the numbers of invitation emails and solicited articles submitted and accepted in *Proceedings B* between September 2017 and January 2021. (*b*) Comparison between the proportion of accepted versus rejected papers between solicited ($N = 96$) and traditional ($N = 11\,583$) submissions in the 2017–2020 period. A significantly higher proportion of solicited vs. traditional-route manuscripts were accepted for publication (see text for details). (Online version in colour.)

across the 30 topics covered by *Proceedings B* (electronic supplementary material, table S1) to find that there was not a significant difference in the topical representation of solicited articles versus articles submitted via the 'traditional' (non-solicited) route from 2018–2020 ($Z = 1.184$, $p = 0.236$). This result finds additional support from a Kendall's rank sum correlation analysis (as implemented in IBM SPSS Statistics v. 27.0) demonstrating a strong positive correlation between the representation of topics covered by the two paper categories ($T = 0.546$, $p < 0.001$). Nevertheless, a visual comparison of topical representation for solicited versus traditional-route papers does hint that the solicited papers might represent a broader swath of biology (figure 3). For example, in the 'Bioengineering' and 'Synthetic Biology' categories solicited submissions made up 2% and 1% of total submissions, respectively (figure 3). By contrast, distinctly less than 1% of papers were submitted in these categories via the 'traditional route'. Similarly, while less than 5% of the 11 583 papers in the traditional route were categorized as Computational Biology, Microbiology, Genomics, Immunology, or Biophysics, greater than 5% of submitted solicited preprints were identified as belonging to one or more of these categories (figure 3). It will be illuminating to repeat these comparisons in a few years when we have a larger set of submitted solicited preprints to include in the analyses.

We performed a similar set of analyses from the same dataset of traditionally submitted manuscripts versus solicited preprints aimed at comparing geographic diversity of the affiliation of the submitting author in these two groups of submissions (electronic supplementary material, table S1). A Wilcoxon signed-ranks test did not reveal any evidence for a difference in geographic representation of authors of submitted papers between traditional versus solicited categories across the 20 countries that contributed at least 1% of all submissions in at least one year between 2018 and 2020 and the pooled submitting authors from all other countries ($Z = 0.122$,

$p = 0.903$). The similarity in geographic representation of authors from these two submission categories is supported by a Kendall's rank correlation analysis, which revealed a strong and positive correlation between the per cent contributions of countries of origin for the submitting author for traditional versus solicited manuscripts ($T = 0.796$, $p < 0.001$). In contrast with topical diversity, visual inspection of the data did not indicate any trends towards any differences. Indeed, there were no countries uniquely represented among co-authors of solicited submissions. Instead, this figure presents a picture of striking similarity between the geographical representation of authors of papers submitted via each of the two routes (figure 4). As for topical diversity, repeating these analyses in a few years when more data from solicited papers are available will provide a more rigorous means of assessing the impact of preprint solicitation on increasing the geographical representation of authors of papers submitted to *Proceedings B*. While the software that we use in our online submission system cannot generate reports including the country of origin of all (versus just submitting) authors, precluding an analysis that included these data, the ability to perform such an analysis in the future could also be illuminating.

## 4. Voices of preprint team members

Twenty-seven of our current and previous members graciously shared their experiences (electronic supplementary material, table S3) in response to a set of eight focused questions and four more open-ended comment-format questions regarding their participation on the Preprint Team (table 1). These early-career scientists are doing a tremendous job as team members while receiving a unique and hands-on chance to be directly involved in science publishing. We used this admittedly small and informal survey from a diverse but not globally representative set of people to take

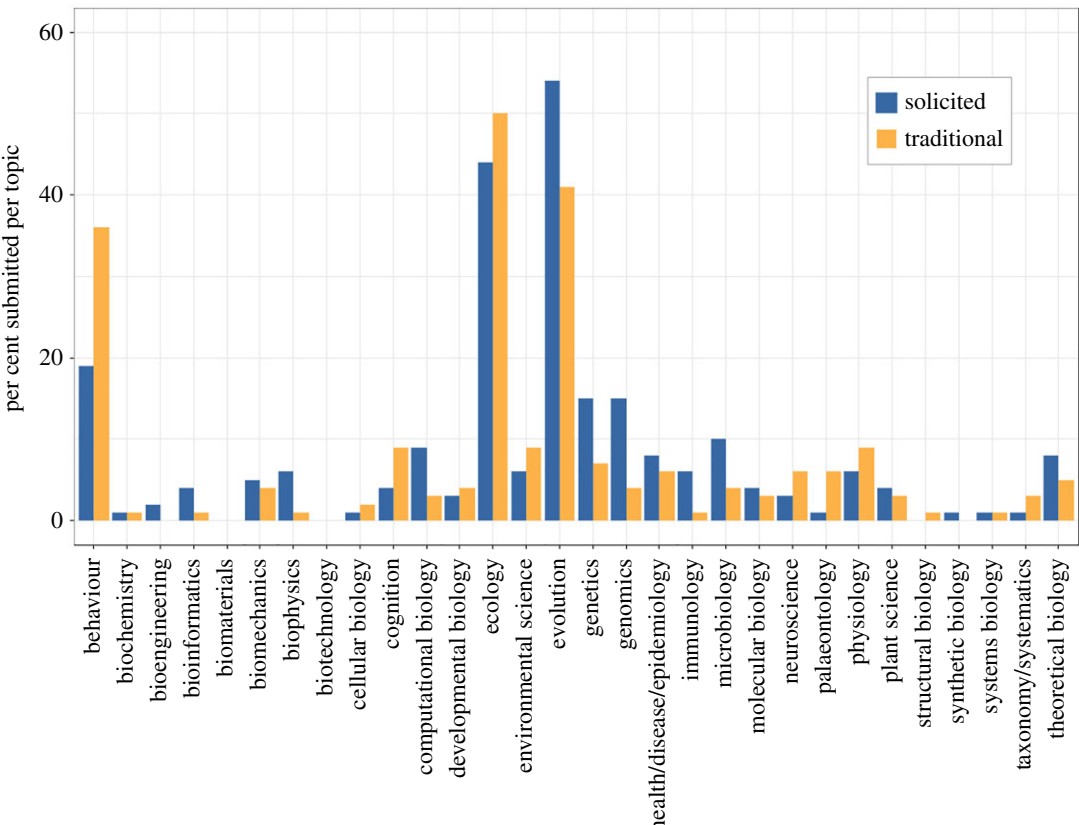

**Figure 3.** Percentage of traditional ($N = 11\,583$) and solicited ($N = 96$) submissions across topical areas covered by *Proceedings B*. Total sums for each type are greater than 100% because authors can choose greater than one topic category per paper. (Online version in colour.)

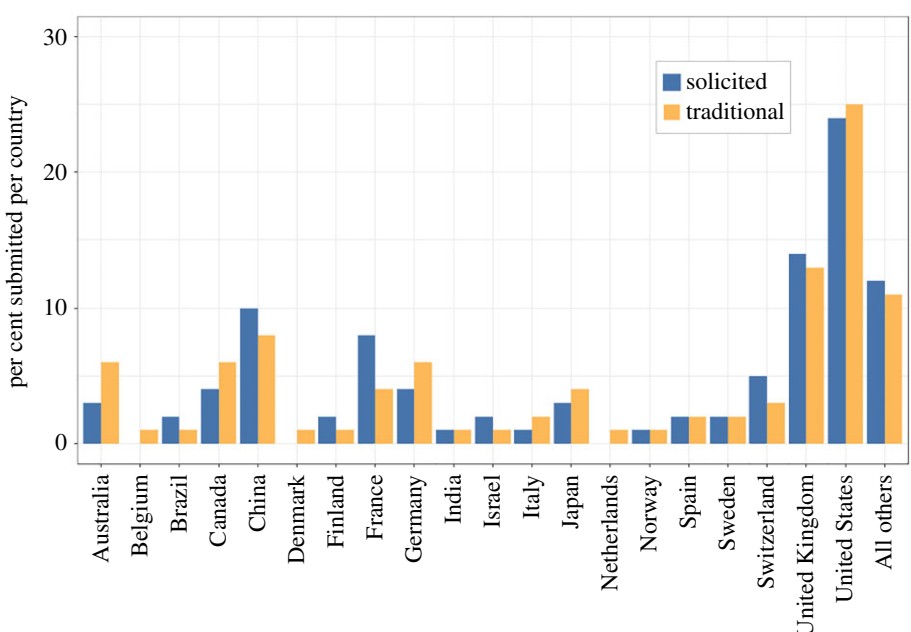

**Figure 4.** Percentage of submitting authors from traditional ($N = 11\,583$) and solicited ($N = 96$) submissions across 20 countries all represented at 1% or greater of traditional submissions along with a pooled category for authors from countries outside of these 20. (Online version in colour.)

initial steps towards characterizing important elements of team member characteristics and experience like career stage and workload (figure 5). We also used this survey to provide some qualitative perceptions of how participation on the Preprint Editorial Team has influenced team member careers.

With respect to the latter, respondents generally agreed (15/27) that participation in the preprint group has improved their academic career, and overwhelmingly (24/27) indicated their work with the group aided their awareness of their research area. Several commented that regular exposure to preprints helped them to stay on top of new developments and trends in their field or to identify papers relevant to their work. Several respondents felt that engaging in the regular review of preprints made them more aware of key aspects of the publishing process such as the manuscript review process, editorial handling, and the scope or 'fit' of research journals.

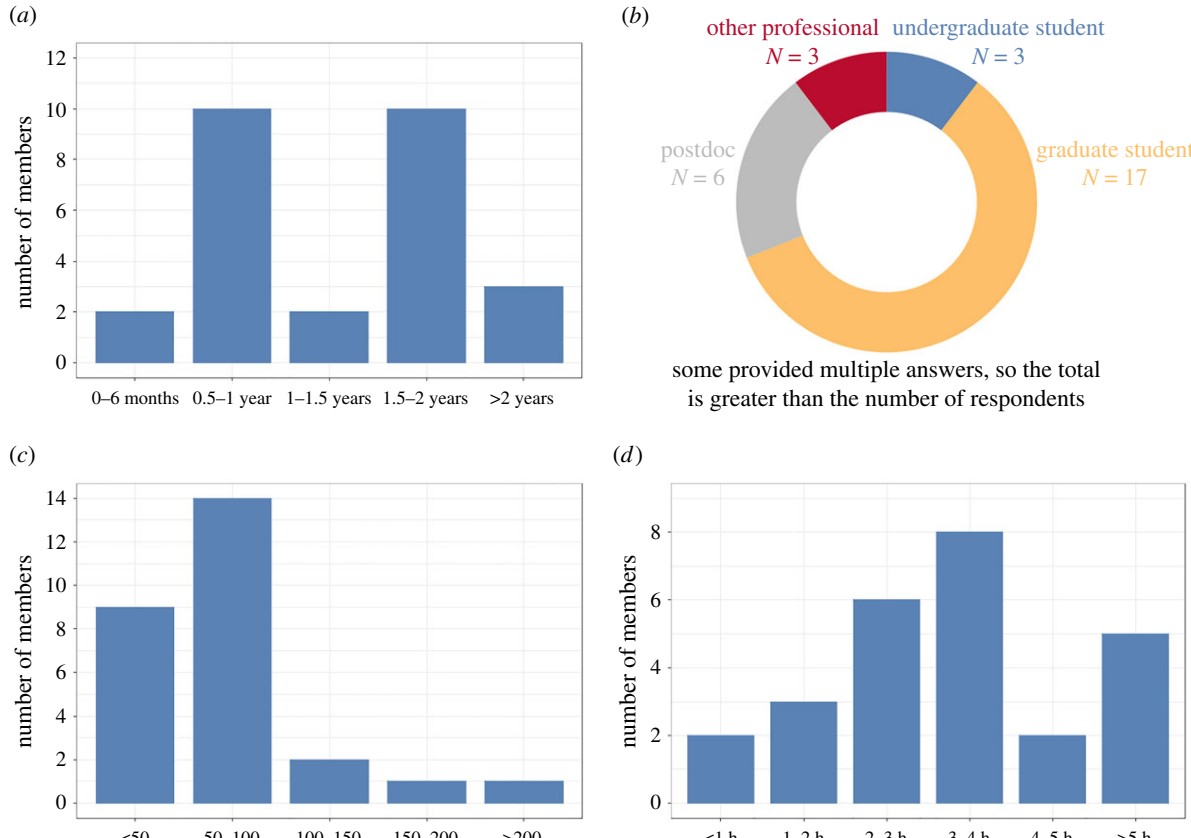

**Figure 5.** Our Preprint Editorial Team members at a glance. (*a*) Length of membership: although members come and go as their careers advance and new responsibilities arise, many participants are with us for a year or more. (*b*) Career stage: the bulk of our members are graduate students, some of whom continue to work with us after accepting postdoctoral or other professional positions. (*c*) Preprints handled: the number of preprints submitted to *bioRχiv* varies by month, and so does the number of preprints handled per team member. (*d*) Time commitment: although we provide a general framework for assessing preprints, each team and each member has their own strategy for assessing preprints. Regardless of strategy, members typically spend 2–5 h per month handling their assigned preprints. (Online version in colour.)

**Table 1.** The survey questions asked of the Preprint Editorial Team members.

| question | abbreviation | long form |
| --- | --- | --- |
| 1 | tenure | How long have you been/were you a member of the group? |
| 2 | career stage | While a member of the group, you were a(n): |
| 3 | preprints handled | How many preprints do you handle in a typical month? |
| 4 | time spent | How many hours do you spend handling preprints in a typical month? |
| 5 | view on career | Participating in this group has helped my academic career. |
| 6 | view on field awareness | Participating in this group has helped my awareness of the state of science/of my field. |
| 7 | preprint likelihood | Participating in this group has made me more likely to deposit my work on a preprint server. |
| 8 | preprint quality | Participating in this group has _________________ my perception of preprint quality. |
| 9 | comments | How has participating in this group helped your academic career? |
| 10 | comments | How has participating in this group impacted your awareness of the publishing process? |
| 11 | comments | How has participating in this group impacted your opinions of preprints? |
| 12 | comments | Is there anything else you would share about your experiences with the group? |

Participants generally indicated their work in the group improved (11/27) or did not change (13/27) their existing viewpoint on the quality of preprints; as well as either no impact (8/27) or an increased likelihood (17/27) to deposit their own work on preprint servers. There were a range of comments on the overall concept of preprints, including notes of their importance in making science accessible to scientists and non-scientists alike, and expressing the view that engagement with and amplification of preprints provides a community service. Altogether, we believe that these survey results reflect both that these early-career scientists are playing a central role in preprint solicitation and that the participation of these scientists is likely to translate into downstream benefits from the establishment of connections

to other scholars to a better understanding of the process of academic publication.

## 5. Challenges: some solutions, but some continue

While the reception of our preprint solicitations has generally been positive, we have also encountered some challenges. One early issue arose when, after sending a non-personalized email to a list of authors, we realized that many of these authors assumed that our solicitation was spam from a predatory journal. We addressed this issue by enlisting a talented then-undergraduate member of the team, Jorge Moreno, to design a Python script that we could pair with a spreadsheet to quickly send out personalized emails. Although this has not been a perfect solution—we still get the occasional annoyed reply from an author who asks us to not email them again—we do think it is a step in the right direction.

An awkward situation emerged on a handful of occasions when we inadvertently solicited papers that had already been solicited by (and not submitted) or submitted to but then rejected by *Proceedings B*. In the former case, we were embarrassed that we had repeatedly emailed authors. In the second case, the understandably frustrated authors would email us, confused that their article was solicited post-rejection. The solution to both problems came from simple computational fixes. First, to avoid repeated solicitations, we added a script to our shared spreadsheet that flagged email/title combinations that we had previously emailed. Second, to avoid soliciting already rejected papers, we used a simple script in Microsoft Excel software to check the titles of a list of rejected papers that we receive every week from the editorial staff at *Proceedings B* against of list of to-be-solicited papers. We then removed any papers that had already been considered and rejected before sending out our solicitation emails each month.

A more nuanced issue is posed by the occasional rejection without review of solicited papers by Associate Editors because of their perception that the paper is 'outside of scope'. This problem is somewhat circular in nature: if our goal is to broaden the scope of *Proceedings B*, we might not yet have appropriate breadth in the Editorial Board, or familiarity with certain topics, to be able to fairly evaluate papers that do provide an expansion in scope. One partial fix to this problem came by simply sending papers back to Editor Neiman that an Associate Editor had rejected without review for one last look. In most of these cases, Neiman agreed with the Associate Editor's decision.

For a few papers, Editor-in-Chief Spencer Barrett and Editor Neiman decided that it was reasonable to send the paper to a different Associate Editor for reconsideration. After a year or so into the preprint solicitation endeavour, we realized that a particular subset of some of the papers that were rejected without review focused on science studies. This was an area of interest for *Proceedings B*, so Editor-in-Chief Barrett, Editor Neiman, and Publishing Editor (and co-author) Shalene Singh-Shepherd together decided to design a new science studies-focused topical category, *Biological Science Practices*. We recruited a science studies scholar, Dr Stephanie Meirmans, to serve as Associate Editor. This new paper category has been quite successful, with 30 submissions—and five papers published—to date since late 2019, including several (e.g. [14]) receiving quite a bit of positive international media attention.

## 6. Summary and Conclusion

We here describe our process for using a widely available and prominent preprint server, *bioRχiv*, to expand access to and scope of *Proceedings B*, a broadly oriented biology journal. Our model has been successful enough to be adopted by other journals (e.g. *Open Biology*), who are using a virtually identical approach to identify and solicit promising preprints for submission. We used comparisons of the focal topics of papers submitted via the traditional (non-solicited) route versus papers solicited via our preprint team to provide a preliminary indication that we might be succeeding in terms of increasing the topical diversity of papers submitted to *Proceedings B*. We also detail the involvement of a large group of early-career researchers and then use a small informal survey to assess workload and how participation affects perception of preprints and career development. This survey, though representing a relatively small sample of team members, revealed generally positive consequences of participation for these members of the preprint team. Altogether, we believe that this paper reflects an overtly successful but still imperfect process for leveraging preprint servers to raise awareness of a journal with a global reach and to move towards achieving goals regarding broader disciplinary coverage and author representation.

Data accessibility. The data are provided in the electronic supplementary material [15].

Authors' Contributions. M.N. and S.S.S. conceived the paper idea; S.S.S. organized and made available the data on submissions and acceptances; M.N. analysed the submission and acceptance data; R.K.B. suggested, designed and executed the survey, and analysed the survey data, and D.P. designed and created the figures. M.N. wrote the manuscript. All authors revised the manuscript, gave final approval for publication, and agree to be held accountable for the work performed therein.

Competing interests. We declare a competing interest; authors Neiman and Singh-Shepherd are on the *Proceedings B* Editorial Board.

Funding. We received no funding for this study.

Acknowledgements. We gratefully acknowledge Editor-in-Chief Spencer Barrett for the suggestion and encouragement for this paper idea and Jorge Moreno and Bennett Brown for Python and Excel scripts critical to our success. Most importantly, we want to express gratitude for the early-career scientists who make up the Preprint Editorial Team, now and in the past. We couldn't do this without you. Several anonymous reviewers also provided helpful feedback that substantially improved the manuscript.

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
