## [Peer Review File · Proceedings of the Royal Society B: Biological Sciences]

Review History

RSPB-2021-0864.R0 (Original submission)

Review form: Reviewer 1

Recommendation

Major revision is needed (please make suggestions in comments)

Scientific importance: Is the manuscript an original and important contribution to its field?

Acceptable

General interest: Is the paper of sufficient general interest?

Acceptable

Quality of the paper: Is the overall quality of the paper suitable?

Marginal

Is the length of the paper justified?

No

Should the paper be seen by a specialist statistical reviewer?

No

Do you have any concerns about statistical analyses in this paper? If so, please specify them explicitly in your report.

No

It is a condition of publication that authors make their supporting data, code and materials available - either as supplementary material or hosted in an external repository. Please rate, if applicable, the supporting data on the following criteria.

Is it accessible?

Yes

Is it clear?

No

Is it adequate?

Yes

Do you have any ethical concerns with this paper?

No

Comments to the Author

It is clear that pre-prints increasingly form an important component of the publishing landscape. For this reason, Proceedings B is wise to give the opportunity careful thought, and to make their reasoning transparent.

This paper was sent for review as a standard contribution to the journal, yet it reads more like an editorial. It contains phrases such as:

‘This opportunity resonated with co-author Neiman, a newly appointed Associate Editor at Proceedings B, for two reasons. First, she was increasingly focused on how science might be applied to enact positive change. Second, Neiman’s leadership of a science-writing course at the University of Iowa had pushed her to tackle the intersections between writing, publishing and reviewing practices, and diversity, equity, and inclusion-related issues like open access, paywalls, and author and reviewer anonymity.’

‘She was enthusiastic about the potential of preprints to increase the accessibility of science and to counter some of the challenges associated with peer review and publication.’

and comes across as too focussed on one person’s aspirations. Surely the aspirations of this exercise should be attributed to the Editorial Board as a whole? I am also uncomfortable with the first author of a paper using a paper to praise their actions.

The idea of using preprints to solicit submissions to the journal is a good one, given appropriate safeguards. I found this paper very vague about the criteria used to identify a ‘promising’ paper, and it was also unclear to me how bias in favour of, or against, a particular piece of work is avoided. For example, is there a tendency to seek out papers on the current ‘hot topic’ or ignore papers on subjects deemed to be unfashionable or contributed by rival groups? I expect that, in practice, the team was scrupulous in this regard, but the important point is that these criteria need to articulated.

The sample sizes for ‘the voices of the preprint team’ are rather small, yet discussion of this represents a substantial portion of text.

The preprint editorial team seems (perhaps by necessity) to be geographically concentrated, so again there may be questions about potential bias, recruitment etc.

Indeed, generally speaking, the text could be much more succinct, and would be better for this.

Two of the Figures lack Figure numbers, and Figure 2 needs to show the n.

The comment about solicited papers representing a broader swathe of biology is not well supported by Figure 2, and the numbers given in text are in any case small sample statistics.

Finally, the abstract asserts that: 'From a social justice perspective, preprints provide a legitimate and effective mechanism to increase scholarly productivity, and especially for those most likely to experience bias or inequity during the publication process'. This is an important statement but what is the evidence that preprints achieve this important goal? I had a look at the Cobb and Penfield & Polka papers that are cited, and while they provide an overview of the preprint field, I wasn't convinced that they provide evidence for progress towards social justice - though the aspiration to support it is mentioned.

Review form: Reviewer 2 (Sean Burns)

Recommendation

Accept with minor revision (please list in comments)

Scientific importance: Is the manuscript an original and important contribution to its field?

Excellent

General interest: Is the paper of sufficient general interest?

Excellent

Quality of the paper: Is the overall quality of the paper suitable?

Excellent

Is the length of the paper justified?

Yes

Should the paper be seen by a specialist statistical reviewer?

No

Do you have any concerns about statistical analyses in this paper? If so, please specify them explicitly in your report.

No

It is a condition of publication that authors make their supporting data, code and materials available - either as supplementary material or hosted in an external repository. Please rate, if applicable, the supporting data on the following criteria.

Is it accessible?

Yes

Is it clear?

No

Is it adequate?

N/A

Do you have any ethical concerns with this paper?

No

Comments to the Author

The authors report on a process to identify and solicit papers from bioRxiv in order to expand the scope of Proceedings B. The process also recognizes the increasing importance of preprints in the scholarly publication and open science processes. Although not stated here, this recognition by Proceedings B is important since studies show that journals can strongly influence author behavior in such matters. A previous colleague, Youngseek Kim (<https://orcid.org/000.0-0001-6542-0792>), and others have published on this topic. In line with this, it's great when journals are transparent about their activities, and this is helpful as such in motivating more participation in open science/access.

One shortcoming with the paper is that the authors highlight how preprints have a social justice aspect to them. However, the authors do not address how their incorporation of preprints into the scholarly cycle of Proceedings B might reduce, e.g., "experience bias or inequity during the publication process." It should not take much for the authors to address this in the final parts of the paper if for not any other reason but to close the loop, so to speak. That is, if the authors do not have more specific data to show how bias or inequities are addressed here, my hope is they can at least address how it might in a more specific way in the future. For example, has this process not only expanded the breadth and scope of Proceedings B, but has the journal also seen more representation from institutions and nations that are often under represented in Proceedings B? There's no reason to cite this paper here, but this reviewer has published on a topic like this and I bring it up only to show how scholarly publishing can mirror societal inequity:

Burns, C.S., & Fox, C.W. (2017). Language and socioeconomics predict geographic variation in peer review outcomes at an ecology journal. *Scientometrics*, 113(2), 1113-1127.
doi:<https://doi.org/10.1007/s11192-017-2517-5>. Open access copy:
<https://works.bepress.com/cseanburns/42/>

One particularly noteworthy aspect of this paper is the inclusion of early career and student scientists in the process of identifying preprints on bioRxiv and in the publication process of Proceedings B, and in how this has expanded to include more people from more institutions and countries. It certainly benefits early career scientists to see this process in closer detail, since publishing has long been like a black box. I think the authors also have a chance to show the social justice aspect of this, too, in that including these people might help open up the publication process more to them and offer paths to their own publishing success and future editorial leadership positions.

In light of the above two comments, I think it would be helpful if the process of identifying and soliciting preprints was accompanied by a mission statement that went beyond increasing the breadth of papers submitted to Proceedings B to include a statement on the social justice aspect of this process. Perhaps this is something the editorial team could discuss in a future meeting.

Finally, it's not clear to this reviewer if this paper is classified as an editorial or a research paper. I think it's a great editorial, but I would ask for more if it was a research paper.

I enjoyed reading it, and applaud Proceedings B for adding this to their process and for including early career, etc people in that process.

Review form: Reviewer 3

Recommendation

Accept with minor revision (please list in comments)

Scientific importance: Is the manuscript an original and important contribution to its field?

Good

General interest: Is the paper of sufficient general interest?

Good

Quality of the paper: Is the overall quality of the paper suitable?

Good

Is the length of the paper justified?

Yes

Should the paper be seen by a specialist statistical reviewer?

No

Do you have any concerns about statistical analyses in this paper? If so, please specify them explicitly in your report.

Yes

It is a condition of publication that authors make their supporting data, code and materials available - either as supplementary material or hosted in an external repository. Please rate, if applicable, the supporting data on the following criteria.

Is it accessible?

Yes

Is it clear?

Yes

Is it adequate?

Yes

Do you have any ethical concerns with this paper?

Yes

Comments to the Author

This is an interesting article about a project that has worked well and should be. The following are points and questions that arose during reading, some of which could probably generate minor improvements, in sequence of the text, rather than in terms of importance:

- The project has two aims, both admirable and interesting: to assess effectiveness of the preprint solicitation process for the journal's benefit and to offer a useful training experience for young researchers. To some extent these could be in conflict if the recommendation is to continue the scheme, in that to be used fairly, there probably needs to be a slightly higher level of quality control over the recommendations made by the volunteers (see below). Some addition comment on this would be interesting (the text says the volunteers are 'fantastic' which is good, but need some checking
- Para beginning line 116. According to what process were subject areas of articles assigned to students? Was there any training or standardised set of criteria to be used for the assessment of abstracts? Was there any editorial oversight at this stage. Neiman checked articles that were selected for solicitation, but was there any (spot) checking of those that were rejected?

- There could usefully be a table or probably a figure in the main body of the article rather than in the supplementary material, summarising succinctly the statistics on number of articles reviewed, number accepted for invitation, number giving various responses etc to the point of acceptance/rejection by the journal. Most of these figures come in the text, but this way readers can see them all at a glance.
- Line 185 - 195. Give full results for Wilcoxon test. It might be worth looking to see if any of the categories used for these analyses could be combined without undue loss of information (e.g. bioengineering and biophysics). Would it be worth using something like a 'biodiversity index' or papers submitted through the two routes?
- Line 198. It would be useful to have a table in the text briefly summarising the questions the student reviewers were asked. Readers will want to know what these are without having to break off and look at supplementary material, if this reader is anything to go by.
- Throughout, the discursive/reflective style is appropriate for this kind of article and makes it easy to read. However, overall, it is a bit wordy and in places repetitive and could be improved by some slight tightening up.

Decision letter (RSPB-2021-0864.R0)

21-May-2021

Dear Dr Neiman:

I am writing to inform you that your manuscript RSPB-2021-0864 entitled "Development, Implementation, and Impact of a New Preprint Solicitation Process at Proceedings B" has, in its current form, been rejected for publication in Proceedings B.

This action has been taken on the advice of referees, who have recommended that substantial revisions are necessary. With this in mind we would be happy to consider a resubmission, provided the comments of the referees are fully addressed. However please note that this is not a provisional acceptance.

Sincerely,
 Professor Gary Carvalho
 Editor, Proceedings B
 mailto: proceedingsb@royalsociety.org

Editor Comments

Comments to Author:

Thank you for submitting your manuscript consideration as a Biological Sciences Practices manuscript. I apologise about the delay in getting the decision to you, though it took some time to command an appropriate and representative range of opinions.

Overall, you will see that the reviewers certainly applaud the idea of such an article as well as the overall approach, though there is a lack of clarity, on several substantive points. In particular, and I agree that while some level of reflection can be engaging and informative, there is a level of over repetition, and over personalisation in terms of individuals concerned, that could be effectively remedied by a change of emphasis and tone. Additionally, and this was identified by referees as a key point in relation to the pre-print approach to provide a more convincing justification and appropriate illustrative detail, on the social justice element of claims made. The rather small sample size of the "voices of the preprint team" need some additional support in terms of representation, as well as some comment concerning the appropriate level of consistency in quality control. I am pleased that the referees have provided sufficient and illustrative detail in their comments and suggestions, that will I hope facilitate the appropriate reconsideration of content and tone. We look forward to receiving a resubmitted manuscript that will enter the peer review process once again. Thank you again for your patience.

Reviewer(s)' Comments to Author:

Referee: 1

Comments to the Author(s)

It is clear that pre-prints increasingly form an important component of the publishing landscape. For this reason, Proceedings B is wise to give the opportunity careful thought, and to make their reasoning transparent.

This paper was sent for review as a standard contribution to the journal, yet it reads more like an editorial. It contains phrases such as:

'This opportunity resonated with co-author Neiman, a newly appointed Associate Editor at Proceedings B, for two reasons. First, she was increasingly focused on how science might be applied to enact positive change. Second, Neiman's leadership of a science-writing course at the University of Iowa had pushed her to tackle the intersections between writing, publishing and reviewing practices, and diversity, equity, and inclusion-related issues like open access, paywalls, and author and reviewer anonymity.'

'She was enthusiastic about the potential of preprints to increase the accessibility of science and to counter some of the challenges associated with peer review and publication.'

and comes across as too focussed on one person's aspirations. Surely the aspirations of this exercise should be attributed to the Editorial Board as a whole? I am also uncomfortable with the first author of a paper using a paper to praise their actions.

The idea of using preprints to solicit submissions to the journal is a good one, given appropriate safeguards. I found this paper very vague about the criteria used to identify a 'promising' paper, and it was also unclear to me how bias in favour of, or against, a particular piece of work is avoided. For example, is there a tendency to seek out papers on the current 'hot topic' or ignore papers on subjects deemed to be unfashionable or contributed by rival groups? I expect that, in

practice, the team was scrupulous in this regard, but the important point is that these criteria need to be articulated.

The sample sizes for 'the voices of the preprint team' are rather small, yet discussion of this represents a substantial portion of text.

The preprint editorial team seems (perhaps by necessity) to be geographically concentrated, so again there may be questions about potential bias, recruitment etc.

Indeed, generally speaking, the text could be much more succinct, and would be better for this.

Two of the Figures lack Figure numbers, and Figure 2 needs to show the n.

The comment about solicited papers representing a broader swathe of biology is not well supported by Figure 2, and the numbers given in text are in any case small sample statistics.

Finally, the abstract asserts that: 'From a social justice perspective, preprints provide a legitimate and effective mechanism to increase scholarly productivity, and especially for those most likely to experience bias or inequity during the publication process'. This is an important statement but what is the evidence that preprints achieve this important goal? I had a look at the Cobb and Penfield & Polka papers that are cited, and while they provide an overview of the preprint field, I wasn't convinced that they provide evidence for progress towards social justice – though the aspiration to support it is mentioned.

Referee: 2

Comments to the Author(s)

The authors report on a process to identify and solicit papers from bioRxiv in order to expand the scope of Proceedings B. The process also recognizes the increasing importance of preprints in the scholarly publication and open science processes. Although not stated here, this recognition by Proceedings B is important since studies show that journals can strongly influence author behavior in such matters. A previous colleague, Youngseek Kim (<https://orcid.org/000.0-0001-6542-0792>), and others have published on this topic. In line with this, it's great when journals are transparent about their activities, and this is helpful as such in motivating more participation in open science/access.

One shortcoming with the paper is that the authors highlight how preprints have a social justice aspect to them. However, the authors do not address how their incorporation of preprints into the scholarly cycle of Proceedings B might reduce, e.g., "experience bias or inequity during the publication process." It should not take much for the authors to address this in the final parts of the paper if for not any other reason but to close the loop, so to speak. That is, if the authors do not have more specific data to show how bias or inequities are addressed here, my hope is they can at least address how it might in a more specific way in the future. For example, has this process not only expanded the breadth and scope of Proceedings B, but has the journal also seen more representation from institutions and nations that are often under represented in Proceedings B? There's no reason to cite this paper here, but this reviewer has published on a topic like this and I bring it up only to show how scholarly publishing can mirror societal inequity:

Burns, C.S., & Fox, C.W. (2017). Language and socioeconomics predict geographic variation in peer review outcomes at an ecology journal. *Scientometrics*, 113(2), 1113-1127.
doi:<https://doi.org/10.1007/s11192-017-2517-5>. Open access copy:
<https://works.bepress.com/cseanburns/42/>

One particularly noteworthy aspect of this paper is the inclusion of early career and student scientists in the process of identifying preprints on bioRxiv and in the publication process of

Proceedings B, and in how this has expanded to include more people from more institutions and countries. It certainly benefits early career scientists to see this process in closer detail, since publishing has long been like a black box. I think the authors also have a chance to show the social justice aspect of this, too, in that including these people might help open up the publication process more to them and offer paths to their own publishing success and future editorial leadership positions.

In light of the above two comments, I think it would be helpful if the process of identifying and soliciting preprints was accompanied by a mission statement that went beyond increasing the breadth of papers submitted to Proceedings B to include a statement on the social justice aspect of this process. Perhaps this is something the editorial team could discuss in a future meeting.

Finally, it's not clear to this reviewer if this paper is classified as an editorial or a research paper. I think it's a great editorial, but I would ask for more if it was a research paper.

I enjoyed reading it, and applaud Proceedings B for adding this to their process and for including early career, etc people in that process.

Referee: 3

Comments to the Author(s)

This is an interesting article about a project that has worked well and should be. The following are points and questions that arose during reading, some of which could probably generate minor improvements, in sequence of the text, rather than in terms of importance:

- The project has two aims, both admirable and interesting: to assess effectiveness of the preprint solicitation process for the journal's benefit and to offer a useful training experience for young researchers. To some extent these could be in conflict if the recommendation is to continue the scheme, in that to be used fairly, there probably needs to be a slightly higher level of quality control over the recommendations made by the volunteers (see below). Some additional comment on this would be interesting (the text says the volunteers are 'fantastic' which is good, but need some checking)
- Para beginning line 116. According to what process were subject areas of articles assigned to students? Was there any training or standardised set of criteria to be used for the assessment of abstracts? Was there any editorial oversight at this stage. Neiman checked articles that were selected for solicitation, but was there any (spot) checking of those that were rejected?
- There could usefully be a table or probably a figure in the main body of the article rather than in the supplementary material, summarising succinctly the statistics on number of articles reviewed, number accepted for invitation, number giving various responses etc to the point of acceptance/rejection by the journal. Most of these figures come in the text, but this way readers can see them all at a glance.
- Line 185 - 195. Give full results for Wilcoxon test. It might be worth looking to see if any of the categories used for these analyses could be combined without undue loss of information (e.g. bioengineering and biophysics). Would it be worth using something like a 'biodiversity index' or papers submitted through the two routes?
- Line 198. It would be useful to have a table in the text briefly summarising the questions the student reviewers were asked. Readers will want to know what these are without having to break off and look at supplementary material, if this reader is anything to go by.
- Throughout, the discursive/reflective style is appropriate for this kind of article and makes it easy to read. However, overall, it is a bit wordy and in places repetitive and could be improved by some slight tightening up.

Author's Response to Decision Letter for (RSPB-2021-0864.R0)

See Appendix A.

RSPB-2021-1248.R0

Review form: Reviewer 2 (Sean Burns)

Recommendation

Major revision is needed (please make suggestions in comments)

Scientific importance: Is the manuscript an original and important contribution to its field?

Acceptable

General interest: Is the paper of sufficient general interest?

Excellent

Quality of the paper: Is the overall quality of the paper suitable?

Good

Is the length of the paper justified?

Yes

Should the paper be seen by a specialist statistical reviewer?

No

Do you have any concerns about statistical analyses in this paper? If so, please specify them explicitly in your report.

Yes

It is a condition of publication that authors make their supporting data, code and materials available - either as supplementary material or hosted in an external repository. Please rate, if applicable, the supporting data on the following criteria.

Is it accessible?

Yes

Is it clear?

Yes

Is it adequate?

Yes

Do you have any ethical concerns with this paper?

Yes

Comments to the Author

Please see the attached document for my review. (See Appendix B)

Decision letter (RSPB-2021-1248.R0)

14-Jun-2021

Dear Dr Neiman:

Your manuscript has now been peer reviewed and the reviews have been assessed by an Associate Editor. The reviewers' comments (not including confidential comments to the Editor) and the comments from the Associate Editor are included at the end of this email for your reference. As you will see, the reviewers and the Editors have raised some concerns with your manuscript and we would like to invite you to revise your manuscript to address them.

I very much appreciate the additional work carried out in the resubmission, and the helpful and easy to follow uploaded response letter. The manuscript has been seen by one of the original reviewers, and I am pleased that in general, there is a consensus that the manuscript has been substantially improved. I have also noted your responses in relation to some of my overall generic comments, which I am also satisfied with. However, you will see that the reviewer has raised a couple of additional points, relating to a more robust and transparent literature evidence base, and the statistical analysis to compare categories of published manuscripts. I will not elaborate further here, since the referee has indeed provided a comprehensive and informative account of not only why there are concerns, but some explicit recommendations of how they may be addressed. I would therefore ask you in the revision, to provide again, a summary of your responses, and a brief overview of the changes made to the manuscript. I would like to send this out once more, to the reviewer, if that reviewer is available and able to look at the manuscript once more. Thank you again for your considered attention.

Research ethics:

Use of animals and field studies:

It is a condition of publication that you make available the data and research materials supporting the results in the article (<https://royalsociety.org/journals/authors/author-guidelines/#data>). Datasets should be deposited in an appropriate publicly available repository and details of the associated accession number, link or DOI to the datasets must be included in the Data Accessibility section of the article (<https://royalsociety.org/journals/ethics-policies/data-sharing-mining/>). Reference(s) to datasets should also be included in the reference list of the article with DOIs (where available).

Please submit a copy of your revised paper within three weeks. If we do not hear from you within this time your manuscript will be rejected. If you are unable to meet this deadline please let us know as soon as possible, as we may be able to grant a short extension.

Best wishes,
Professor Gary Carvalho
mailto: proceedingsb@royalsociety.org

Associate Editor

Comments to Author:

Please see Editor comments embedded in the decision letter above.

Reviewer(s)' Comments to Author:

Referee: 2

Comments to the Author(s).

Please see the attached document for my review.

Author's Response to Decision Letter for (RSPB-2021-1248.R0)

See Appendices C & D.

RSPB-2021-1248.R1 (Revision)

Review form: Reviewer 1 (Sean Burns)

Recommendation

Accept as is

Scientific importance: Is the manuscript an original and important contribution to its field?

Good

General interest: Is the paper of sufficient general interest?

Excellent

Quality of the paper: Is the overall quality of the paper suitable?

Excellent

Is the length of the paper justified?

Yes

Should the paper be seen by a specialist statistical reviewer?

No

Do you have any concerns about statistical analyses in this paper? If so, please specify them explicitly in your report.

No

It is a condition of publication that authors make their supporting data, code and materials available - either as supplementary material or hosted in an external repository. Please rate, if applicable, the supporting data on the following criteria.

Is it accessible?

Yes

Is it clear?

Yes

Is it adequate?

Yes

Do you have any ethical concerns with this paper?

No

Comments to the Author

Thank you for adding a more robust literature review, for revising the statistics, and for providing SPSS output.

Decision letter (RSPB-2021-1248.R1)

23-Jun-2021

Dear Dr Neiman

I am pleased to inform you that your manuscript entitled "Development, Implementation, and Impact of a New Preprint Solicitation Process at Proceedings B" has been accepted for publication in Proceedings B.

Data Accessibility section

Open Access

Your article has been estimated as being 7 pages long. Our Production Office will be able to confirm the exact length at proof stage.

Paper charges

Sincerely,

Professor Gary Carvalho

Appendix A

June 1, 2021

Dear Gary,

Thank you for handling our MS (RSPB-2021-0864). We were very pleased that the MS was generally reviewed favourably, and we found the reviewers' critiques to be very helpful. Our responses to each of their points are embedded below in *italics*; we were able to respond to and accommodate all suggestions and criticisms. We view the revised manuscript to be much improved.

Thank you for your consideration and for your constructive work on behalf of our paper!

Sincerely,
Maurine Neiman, on behalf of the authors

21-May-2021

Dear Dr Neiman:

I am writing to inform you that your manuscript RSPB-2021-0864 entitled "Development, Implementation, and Impact of a New Preprint Solicitation Process at Proceedings B" has, in its current form, been rejected for publication in Proceedings B.

This action has been taken on the advice of referees, who have recommended that substantial revisions are necessary. With this in mind we would be happy to consider a resubmission, provided the comments of the referees are fully addressed. However please note that this is not a provisional acceptance.

Sincerely,

Professor Gary Carvalho
Editor, Proceedings B
mailto:proceedingsb@royalsociety.org

Editor Comments

Comments to Author:

Thank you for submitting your manuscript consideration as a Biological Sciences Practices manuscript. I apologise about the delay in getting the decision to you, though it took some time to command an appropriate and representative range of opinions.

Overall, you will see that the reviewers certainly applaud the idea of such an article as well as the overall approach, though there is a lack of clarity, on several substantive points. In particular, and I agree that while some level of reflection can be engaging and informative, there is a level of over repetition, and over personalisation in terms of individuals concerned, that could be effectively remedied by a change of emphasis and tone. Additionally, and this was identified by referees as a key point in relation to the preprint approach to provide a more convincing justification and appropriate illustrative detail, on the social justice element of claims made. The rather small sample size of the "voices of the preprint team" need some additional support in terms of representation, as well as some comment concerning the appropriate level of consistency in quality control. I am pleased that the referees have provided sufficient and illustrative detail in their comments and suggestions, that will I hope facilitate the appropriate reconsideration of content and tone. We look forward to receiving a resubmitted manuscript that will enter the peer review process once again. Thank you again for your patience.

Thank you, Gary, for the constructive comments and reviews! We really appreciate the feedback, and we find the revised version to be much improved. Indeed, we feel like we have thoroughly and appropriately responded to all of the suggestions. In particular, we have streamlined the manuscript, with special focus on emphasis and tone as recommended. We have also toned down elements of some of the connections we claimed between social justice and preprints, which are reasonable but are indeed not yet (to the extent that we could determine) backed up in literature. We did expand on the social justice benefits experienced by early-career scientist members of the preprint team, provided some caveats regarding the small sample size of our survey, expanded on the details of how we select solicited manuscripts, and added additional analyses in response to the reviewer suggestion regarding geographical diversity.

Reviewer(s)' Comments to Author:

Referee: 1

Comments to the Author(s)

It is clear that pre-prints increasingly form an important component of the publishing landscape. For this reason, Proceedings B is wise to give the opportunity careful thought, and to make their reasoning transparent.

This paper was sent for review as a standard contribution to the journal, yet it reads more like an editorial. It contains phrases such as:

'This opportunity resonated with co-author Neiman, a newly appointed Associate Editor at Proceedings B, for two reasons. First, she was increasingly focused on how science might be applied to enact positive change. Second, Neiman's leadership of a science-writing course at the University of Iowa had pushed her to tackle the intersections between writing, publishing and reviewing practices, and diversity, equity, and inclusion-related issues like open access, paywalls, and author and reviewer anonymity.'

'She was enthusiastic about the potential of preprints to increase the accessibility of science and to counter some of the challenges associated with peer review and publication.'

and comes across as too focussed on one person's aspirations. Surely the aspirations of this exercise should be attributed to the Editorial Board as a whole? I am also uncomfortable with the first author of a paper using a paper to praise their actions.

We as authors also felt a bit awkward about this wording, and were not intending to "praise" our actions per se, but provide a description of the process. The specific aspirations described does provide an accurate reflection of both the thought process of AE Neiman, who volunteered for the role for the reasons that she specified, as well as the role of the editorial board. Nevertheless, we have substantially revised the language in accordance with this critique (throughout, and especially on pages 4-7).

The idea of using preprints to solicit submissions to the journal is a good one, given appropriate safeguards. I found this paper very vague about the criteria used to identify a 'promising' paper, and it was also unclear to me how bias in favour of, or against, a particular piece of work is avoided. For example, is there a tendency to seek out papers on the current 'hot topic' or ignore papers on subjects deemed to be unfashionable or contributed by rival groups? I expect that, in practice, the team was scrupulous in this regard, but the important point is that these criteria need to be articulated.

This is a difficult point to address because the criteria used are indeed holistic and dynamic and might be viewed as rather vague: each team member is individually trained from the start, using text/language similar to the following:

"We recommend that you look over recently published papers in Proceedings B in your area of expertise. You may also want to look at preprints that other members of your team have suggested in the past few months (please note that the entries on the solicitation spreadsheet highlighted in the orange were not solicited, and that Maurine provides specific feedback regarding why the paper wasn't solicited).

What you will find is that both the published articles and the preprints that we have solicited are those that would be of broad interest to biologists on a whole (for example, if it were a genome paper, it might be one that both describes a genome sequence, but also performs some comparative analyses or highlights interesting pathways or genome architectural differences compared to previously published journals). I am an evolutionary biologist, and ask myself when looking at preprint titles/abstracts/etc. if myself and my colleagues who work on, for example, cell biology or developmental biology would think the findings of the preprint were interesting and valuable. Proc B is also relatively high impact, so usually I also assess if the work has some aspect of novelty.

Process-wise, our recommendation would be to scan over the titles first, and rule out any that have a narrow scope or are clearly methodological. Then we would recommend skimming the abstract of those that have interesting and fairly broad titles to get a sense of what questions the paper is asking, and how they present their findings"

Team members also learn as they become more experienced and as other team members and Maurine, Robin, and Dorota provide feedback. We pay no (deliberate) attention whatsoever to "rival groups" or

“hot topics”, instead making a distinct effort to solicit papers that we feel are a good fit for the scope of the journal, but that the authors might not realize as such until they hear from us. We have provided a lot more of this type of context in the revision (pages 4-7).

The sample sizes for ‘the voices of the preprint team’ are rather small, yet discussion of this represents a substantial portion of text.

We did the best we could to solicit as many interviews as possible from our team, but we can only present the responses that we actually received! We do think that these voices warrant an appreciable discussion in the text, so we have left that intact. We do now acknowledge the limitations imposed by the small sample size (line 33, 213-214).

The preprint editorial team seems (perhaps by necessity) to be geographically concentrated, so again there may be questions about potential bias, recruitment etc.

We agree – we recruit worldwide via social media and personal networks, and we make every attempt to include team members from a diversity of regions, but we cannot include people who do not apply. We now acknowledge this limitation in the text (line 214).

Indeed, generally speaking, the text could be much more succinct, and would be better for this.

We have taken this feedback into account as we have worked on our revision, and we view the revised manuscript to be improved in this respect.

Two of the Figures lack Figure numbers, and Figure 2 needs to show the n.

We have fixed these omissions.

The comment about solicited papers representing a broader swathe of biology is not well supported by Figure 2, and the numbers given in text are in any case small sample statistics.

We have toned down this text, adding more qualifiers like “preliminary” and emphasizing the small sample size for the solicited paper side of the comparison (line 29, 165, 179-181, 187-189).

Finally, the abstract asserts that: ‘From a social justice perspective, preprints provide a legitimate and effective mechanism to increase scholarly productivity, and especially for those most likely to experience bias or inequity during the publication process’. This is an important statement but what is the evidence that preprints achieve this important goal? I had a look at the Cobb and Penfield & Polka papers that are cited, and while they provide an overview of the preprint field, I wasn’t convinced that they provide evidence for progress towards social justice – though the aspiration to support it is mentioned.

We agree that this statement needed better support, and we couldn’t find it in the literature. Accordingly, we removed this sentence from the abstract in the revised version.

Referee: 2

Comments to the Author(s)

The authors report on a process to identify and solicit papers from bioRxiv in order to expand the scope

of Proceedings B. The process also recognizes the increasing importance of preprints in the scholarly publication and open science processes. Although not stated here, this recognition by Proceedings B is important since studies show that journals can strongly influence author behavior in such matters. A previous colleague, Youngseek Kim (<https://orcid.org/000.0-0001-6542-0792>), and others have published on this topic. In line with this, it's great when journals are transparent about their activities, and this is helpful as such in motivating more participation in open science/access.

One shortcoming with the paper is that the authors highlight how preprints have a social justice aspect to them. However, the authors do not address how their incorporation of preprints into the scholarly cycle of Proceedings B might reduce, e.g., "experience bias or inequity during the publication process." It should not take much for the authors to address this in the final parts of the paper if for not any other reason but to close the loop, so to speak. That is, if the authors do not have more specific data to show how bias or inequities are addressed here, my hope is they can at least address how it might in a more specific way in the future. For example, has this process not only expanded the breadth and scope of Proceedings B, but has the journal also seen more representation from institutions and nations that are often under represented in Proceedings B? There's no reason to cite this paper here, but this reviewer has published on a topic like this and I bring it up only to show how scholarly publishing can mirror societal inequity:

.
Burns, C.S., & Fox, C.W. (2017). Language and socioeconomic predict geographic variation in peer review outcomes at an ecology journal. *Scientometrics*, 113(2), 1113-1127. doi:<https://doi.org/10.1007/s11192-017-2517-5>. Open access copy: <https://works.bepress.com/cseanburns/42/>

We really appreciate this suggestion, and we have now used available data to compare geographic distribution of authors of papers submitted via the traditional vs. solicited route. While no trends were apparent, we think it is a distinct improvement to be able to address this point, note that analyses in the future will be more powerful, and discuss the negative outcome to date. We have added text in the revised MS that addresses this point (lines 190-206; Fig. 4).

One particularly noteworthy aspect of this paper is the inclusion of early career and student scientists in the process of identifying preprints on bioRxiv and in the publication process of Proceedings B, and in how this has expanded to include more people from more institutions and countries. It certainly benefits early career scientists to see this process in closer detail, since publishing has long been like a black box. I think the authors also have a chance to show the social justice aspect of this, too, in that including these people might help open up the publication process more to them and offer paths to their own publishing success and future editorial leadership positions.

This is a great suggestion, and we have now included mention of these benefits (lines 35-38, 232-235).

In light of the above two comments, I think it would be helpful if the process of identifying and soliciting preprints was accompanied by a mission statement that went beyond increasing the breadth of papers submitted to Proceedings B to include a statement on the social justice aspect of this process. Perhaps this is something the editorial team could discuss in a future meeting.

An excellent suggestion that would indeed require Editorial Board discussion – we will make certain to bring it up at our next annual meeting.

Finally, it's not clear to this reviewer if this paper is classified as an editorial or a research paper. I think it's

a great editorial, but I would ask for more if it was a research paper.

This paper was solicited by the Editor-in-Chief as a research paper. We have added additional analyses and a figure regarding geographical representation, but we will leave it up to editorial discretion to decide how to proceed.

I enjoyed reading it, and applaud Proceedings B for adding this to their process and for including early career, etc people in that process.

We really appreciate the positive feedback!

Referee: 3

Comments to the Author(s)

This is an interesting article about a project that has worked well and should be. The following are points and questions that arose during reading, some of which could probably generate minor improvements, in sequence of the text, rather than in terms of importance:

We appreciate the positive feedback!

- The project has two aims, both admirable and interesting: to assess effectiveness of the preprint solicitation process for the journal's benefit and to offer a useful training experience for young researchers. To some extent these could be in conflict if the recommendation is to continue the scheme, in that to be used fairly, there probably needs to be a slightly higher level of quality control over the recommendations made by the volunteers (see below). Some additional comment on this would be interesting (the text says the volunteers are 'fantastic' which is good, but need some checking

We appreciate the positive feedback and the constructive suggestions. We have implemented them throughout the MS, and especially on pages 4-7.

- Para beginning line 116. According to what process were subject areas of articles assigned to students? Was there any training or standardised set of criteria to be used for the assessment of abstracts? Was there any editorial oversight at this stage. Neiman checked articles that were selected for solicitation, but was there any (spot) checking of those that were rejected?

We have provided several additional paragraphs of context regarding team member training and article selection (pages 4-7).

- There could usefully be a table or probably a figure in the main body of the article rather than in the supplementary material, summarising succinctly the statistics on number of articles reviewed, number accepted for invitation, number giving various responses etc to the point of acceptance/rejection by the journal. Most of these figures come in the text, but this way readers can see them all at a glance.

This is a great suggestion, and we have now added a new figure (Fig. 2) that presents these data.

- Line 185 - 195. Give full results for Wilcoxon test. It might be worth looking to see if any of the categories used for these analyses could be combined without undue loss of information (e.g.

bioengineering and biophysics). Would it be worth using something like a 'biodiversity index' or papers submitted through the two routes?

We have added the Wilcoxon test statistics (lines 179, 196). We also leave the analyses unpooled because the authors select the paper categories while submitting, so we do believe that we will obfuscate or lose useful information if we alter category identity. We are also concerned that category "diversity" might be in the eye of the beholder – some scientists might view some categories to be combinable (e.g., my molecular biology colleagues in my home department think ecology and evolution are the same thing), while others would not.

- Line 198. It would be useful to have a table in the text briefly summarising the questions the student reviewers were asked. Readers will want to know what these are without having to break off and look at supplementary material, if this reader is anything to go by.

Good suggestion! We have added this table (Table 1).

- Throughout, the discursive/reflective style is appropriate for this kind of article and makes it easy to read. However, overall, it is a bit wordy and in places repetitive and could be improved by some slight tightening up.

We agree, and we have made changes with this critique in mind throughout the paper.

.

Appendix B

Dear Editor,

I think the authors have greatly improved this paper and the figures. I think the paper is very well written, that the topic is important, and that it is important that readers of the journal are aware of the efforts to incorporate preprints and early career scientists into the publishing process.

However, although I think the authors made appropriate revisions to the paper based on the feedback provided by the reviewers, in this second round, I have caught some additional issues that I didn't notice in the first review. My sincere apologies to the authors for that. As someone outside the field and thus not part of this journal's core audience, I missed that this paper was submitted as a Biological Science Practices article type, and I had failed to recognize what that means. It has since helped that I was able to read up on that article type for the journal.

In this second round, I think the authors could improve the paper in two ways.

More Robust Literature Review

First, I think the authors could incorporate more literature. The authors could support their motivation by including literature on open access, preprints in biology, and like. See these results for perhaps some helpful suggestions:

- 1) <https://scholar.google.com/scholar?q=preprints+open+access+biology>
- 2) <https://scholar.google.com/scholar?q=open+science+preprints>

Since the authors focus on bioRxiv, I think they could also include literature on that topic, especially as it pertains to the motivation to solicit from this repository:

- 1) <https://scholar.google.com/scholar?q=biorxiv+preprints+journals>

Finally, since the authors involve early career scientists and student scientists in the process of identifying and soliciting manuscripts, I think they could incorporate literature on why it is important for people in the early stages of their careers to be aware of modern day scientific publishing:

- 1) <https://scholar.google.com/scholar?q=early+career+scientists+publishing>
- 2) <https://scholar.google.com/scholar?q=early+career+scientists+preprints>

They may also modify the last two searches above to identify literature that focuses on diversity and inclusion in STEM or in the biological sciences and as it pertains to early career scientists.

Re-review the Statistical Tests

Chi Square Test: Accepted Papers

Second, in this second round of reviews, I noticed a possible error with the Chi-Square test, and thus the remainder of my review is dedicated to the statistical tests the authors used. However, I would also add that even though I have training in some statistics, I am not a professional statistician, and it could be that the issues I found and described below are incorrect.

The authors conducted a chi-square test to determine if there was a difference between the acceptance rates of solicited papers and papers submitted through the traditional route. However, when doing a chi-square test, it's my understanding that one category should not be a subset of another category, and it appears that the authors compare accepted out of submitted for solicited versus non-solicited papers. I believe it is more appropriate to compare accepted versus not-accepted for solicited and non-solicited papers, since accepted and not-accepted are mutually exclusive, which is an assumption of the chi-square test. If that's correct, and if the decision criteria is $p < 0.05$, then there is a statistical difference between acceptance rates for solicited and non-solicited papers:

	Accepted	Not-Accepted	Total
Solicited	29	67	96
Non-Solicited	2445	9138	11583
Total	2474	9208	11679

X-squared = 4.1928, df = 1, p-value = 0.04059

Also, it may not be necessary to apply the Yates correction since this is recommended primarily when cell values are small (e.g., less than 10). If Yates correction is not applied here, then the chi-square statistic is:

X-squared = 4.7221, df = 1, p-value = 0.02978

In my outside reading on this, I found that SPSS may be automatically performing the Yates correction for 2x2 contingency tables. I'm not sure if this is true, but it could be something that the authors check.

Also, the authors could perform a Fisher's Exact Test for Count Data (see link below). Since 30.2% of the solicited papers were accepted but only 21.1% of the non-solicited papers were accepted, the Fisher's test will help highlight the effect size, via an odds ratio, and determine if there is a statistical difference. The test, for me, shows that the odds are statistically higher for solicited papers to be accepted than non-solicited papers, or the odds of acceptance for solicited papers is 1.617 times the odds of acceptance of non-solicited papers. These are my results in R:

Fisher's Exact Test for Count Data

```
data: papers
p-value = 0.03324
alternative hypothesis: true odds ratio is not equal to 1
95 percent confidence interval:
 1.006177 2.542539
sample estimates:
odds ratio
 1.617614
```

More on this here: <https://www.uvm.edu/~statdhtx/StatPages/R/Chi-Square-Folder/chi-square-alternatives.html>

Wilcoxon Signed-Rank Tests: Topical and Geographical Tests

Since I use R and not SPSS, it's my understanding that these two programs use slightly different methods to calculate various statistics, but because of my doubts about the chi-square results reported in the paper, I tested the Wilcoxon tests, too, and received very different numbers from what the authors reported that cannot be explained by the different software. Using the data supplied in "Supplemental Tables 1_2.xlsx" for Tables 1 and 2:

When testing the topics (p. 13) for solicited and non-solicited, R returns:

$Z = -0.84$, $p = 0.400$

To reproduce this, I inputted the topical data from the spreadsheet as a "topical" dataframe in R and ran the following code to run a Wilcoxon signed-ranks test to get the Wilcoxon test and a Z statistic for the test:

```
topicalwilcoxon <- wilcox.test(topical$traditional, topical$solicited, paired = TRUE)
topicalZtest <- qnorm(topicalwilcoxon$p.value/2)
```

```
> topicalwilcoxon
Wilcoxon signed rank test with continuity correction

data: topical$traditional and topical$solicited
V = 142, p-value = 0.4002
alternative hypothesis: true location shift is not equal to 0
```

```
> topicalZtest
[1] -0.8413277
```

I repeated the same process for the geographical results (p. 14). For the difference between solicited and non-solicited, R returns:

$Z = -0.098$, $p = 0.922$

Since these results differ substantially from the authors', I would ask that they double check their work.

However, an alternative to the Wilcoxon tests would be to run a Kendall rank correlation (https://en.wikipedia.org/wiki/Kendall_rank_correlation_coefficient), and I think it might make more intuitive sense to readers. Using the authors' data, I get the following results:

```
Topical Kendall correlation: 0.546
Geographical Kendall correlation: 0.796
```

I think that the moderate correlation of topics of 0.546 makes sense given that the authors note that solicited papers are including topics not normally published by the journal. They write:

"a visual comparison of topical representation for solicited vs. traditional-route papers does hint that the solicited papers might represent a broader swatch of biology" (p. 13).

Also, I think the high correlation of geographies makes sense given that the authors note that there does not seem to be much difference between the geographical locations of authors for solicited and traditionally routed papers. They write:

"In contrast to topical diversity, visual inspection of the [geographical] data did not indicate any trends towards any differences" (p. 14).

Appendix C

*Nonparametric Tests: Related Samples.
 NPTESTS
 /RELATED TEST(PercentReg PercentSol) WILCOXON
 /MISSING SCOPE=ANALYSIS USERMISSING=EXCLUDE
 /CRITERIA ALPHA=0.05 CILEVEL=95.

Nonparametric Tests

[DataSet1] /Users/mneiman/Google Drive/Documents for Google/Papers/Preprint Team for Proc B/preprint data.sav

Hypothesis Test Summary

	Null Hypothesis	Test	Sig. ^{a,b}
1	The median of differences between PercentReg and PercentSol equals 0.	Related-Samples Wilcoxon Signed Rank Test	.236

Hypothesis Test Summary

	Decision
1	Retain the null hypothesis.

- a. The significance level is .050.
- b. Asymptotic significance is displayed.

Related-Samples Wilcoxon Signed Rank Test

PercentReg, PercentSol

Related-Samples Wilcoxon Signed Rank Test Summary

Total N	30
Test Statistic	255.000
Standard Error	43.909
Standardized Test Statistic	1.184
Asymptotic Sig.(2-sided test)	.236

Related-Samples Wilcoxon Signed Rank Test

Continuous Field Information PercentReg

Continuous Field Information PercentSol

Appendix D

GET

FILE= '/Users/mneiman/Google Drive/Documents for Google/Papers/Preprint Team
for Proc B/preprint data geo.sav' .

DATASET NAME DataSet2 WINDOW=FRONT.

*Nonparametric Tests: Related Samples.

NPTESTS

/RELATED TEST(PercentReg PercentSol) WILCOXON

/MISSING SCOPE=ANALYSIS USERMISSING=EXCLUDE

/CRITERIA ALPHA=0.05 CILEVEL=95.

Nonparametric Tests

[DataSet2] /Users/mneiman/Google Drive/Documents for Google/Papers/Preprint Team
for Proc B/preprint data geo.sav

Hypothesis Test Summary

	Null Hypothesis	Test	Sig. ^{a,b}
1	The median of differences between PercentReg and PercentSol equals 0.	Related-Samples Wilcoxon Signed Rank Test	.903

Hypothesis Test Summary

	Decision
1	Retain the null hypothesis.

a. The significance level is .050.

b. Asymptotic significance is displayed.

Related-Samples Wilcoxon Signed Rank Test

PercentReg, PercentSol

Related-Samples Wilcoxon Signed Rank Test Summary

Total N	21
Test Statistic	74.000
Standard Error	20.433
Standardized Test Statistic	-.122
Asymptotic Sig.(2-sided test)	.903

Related-Samples Wilcoxon Signed Rank Test

Continuous Field Information PercentReg

Continuous Field Information PercentSol